# Experimental Research on the Treatment of Stormwater Contaminated by Disinfectants Using Recycled Materials—Hemp Fiber and Ceramzite

**DOI:** 10.3390/ijerph192114486

**Published:** 2022-11-04

**Authors:** Marina Valentukeviciene, Ieva Andriulaityte, Ramune Zurauskiene

**Affiliations:** Environmental Engineering Faculty, Vilnius Gediminas Technical University, LT-10223 Vilnius, Lithuania

**Keywords:** disinfectants, stormwater, pollution indicators, recycled materials

## Abstract

Pollution caused by the use of disinfectants in public spaces is a relatively new form of environmental contamination. During the COVID-19 pandemic of 2020–2021, early research showed a sevenfold increase in the use of disinfectants to clean outdoor spaces and a corresponding increase in environmental pollution. Typically, after entering stormwater systems, disinfectants are carried to surface waters (e.g., rivers, seas and lakes) where they react with various elements to form harmful compounds. In the absence of data, it is not possible to determine accurate levels of pollution according to the latest scientific information. Our enquiry demonstrates that stormwater pollution indicators (pH, conductivity, turbidity and color intensity) change depending on the amounts of disinfectants present. Laboratory tests were conducted using hemp fiber and ceramzite, in which filtered stormwater samples contaminated with different amounts of disinfectants showed decreases in the amounts of active chlorine from 2.93 ppm to 1.0 ppm. Changes in pH levels, conductivity, turbidity and color intensity were monitored before and after filtration; pH indicators changed slightly (from 7.81 to 7.85), turbidity changes varied in the range of 0.070–0.145 NTU and the highest value of color intensity (1.932 AV) was obtained when 50 mL of disinfectant was added to the investigated sample water. This article presents the results of our research into the impact of disinfectants on stormwater. Further investigation is needed in order to determine the impacts of chemical substances on our water ecosystem.

## 1. Introduction

Disinfectants are effective in deactivating viruses and microorganisms.

During the COVID-19 pandemic, which started at the end of 2019, there was an increase in the use of disinfectants to clean outdoor spaces. In an attempt to arrest, and possibly halt, the spread of contagion during the pandemic, streets, highways and other public spaces were sprayed using various chemical substances, which led to increased environmental pollution. According to one estimate, from the start of the pandemic until March 2020, more than 200 tonnes of disinfectants were used in certain regions [1]. Furthermore, during February and March 2020, amounts of chlorine were found to be present in investigated lakes [2].

Similar results were obtained by researchers in Bulgaria, who found that the use of disinfectants to clean public spaces had led to increases in concentrations of disinfectant by-products of up to two to four times in local rivers [3]. Notwithstanding these harmful effects, it is recognized scientifically that disinfection is still the best way to kill viruses [4]. However, intensive disinfectant use produces harmful side effects on the aquatic environment [5,6,7,8]. After entering stormwater, these biocidal substances are transported to surface water bodies where they impact water quality indicators, disrupt biological balances and form dioxins and other harmful compounds (e.g., chloramines, etc.).

Currently, about 250 different chemical substances are classified as biocides by the European Union. About one-third of these chemicals are used as disinfectants to clean surfaces [9]. The suitability of a specific disinfectant for surface cleaning is determined mainly by the active substances it contains. Disinfectants can be classified according to degree of exposure as well as their action, application and chemical structure. The efficacy of disinfection depends on the following parameters: chemical characteristics, temperature, pH, concentration and contact time [9]. Various studies have shown that, during the pandemic, outdoor spaces and surfaces were disinfected mostly with biocides based on organic chlorine compounds (for instance, various hypochlorite solutions). The use of disinfectants, such as sodium hypochlorite (bleach), calcium hypochlorite (bleaching powder), sodium dichloroisocyanurate (NaDCC), chloramine and chlorine dioxide, to fight the SARS-CoV-2 virus increased significantly during the pandemic [10]. For example, if the use of chlorine-based disinfectants reached 19.9% before the pandemic, it rose to 48.7% after the COVID-19 outbreak [11].

One of the most effective and commonly applied disinfectants for surface cleaning is sodium hypochlorite (NaOCl) [12]. The assumption has always been that once hard surfaces have been sprayed with hypochlorite or other chlorine-containing disinfectants, high concentrations of active chlorine remain on the surfaces, which are likely to be transferred to stormwater and surface water bodies via runoff [13].

When chlorine compounds enter stormwater, they react with the organic, inorganic and anthropogenic pollutants present to form a variety of by-products, such as haloacetic acids, trihalomethanes, dibromochloromethane, bromodichloromethane, tribromoethane, dichloroacetonitrile, chloramines and other compounds [14,15]. These toxic compounds affect aquatic ecosystems and may also have various side effects on the water environment [16,17,18]. For example, increased chlorine concentration can cause chlorine toxicity in plants [19] and ecological imbalance [20]. Chlorine by-products accumulate in plankton and fish in the aquatic environment and can cause negative chronic effects in aquatic organisms [21].

Clearly, more studies are needed to evaluate the real impact of chlorine compounds on water ecosystems in order to provide possible methods of treatment. If the requirements for sustainable wastewater management and global trends, such as Agenda 2030 and the Green Deal, are considered, the most effective way to remove disinfectants from stormwater is to apply so-called green solutions. These may include: ecological wetlands, photocatalytic processes, biological decomposition [1], phytoremediation and the removal of pollutants using natural sorbents and different recycled materials generated in the construction sector [22]. Biocoagulants offer a more environmentally and economically effective tool to reduce water turbidity than chemical coagulants [23].

The novelty of this research consists in using the waste materials ceramzite and hemp fiber to treat stormwater in order to reduce disinfectant pollution.

This study presents the results of laboratory experiments performed to evaluate the impact of disinfectants on stormwater pollution indicators and also discusses possible methods of removal using natural sorbents.

## 2. Materials and Methods

The research presented here was conducted at the laboratory facilities of Vilnius Gediminas Technical University, Lithuania. The following materials were used in the experiments:-Test sample collected (20 litres) in the potentially contaminated area at the stormwater outlet (Trakai city, Lithuania);-Disinfectants: didecyldimethylammonium chloride, sodium hypochlorite;-Distilled water, glass jars, 100 mm-diameter glass column;-Recycled materials: ceramzite drainage layer, hemp fiber;-Measuring equipment to identify stormwater pollution indicators (pH, conductivity, turbidity, color, active chlorine).

The research was conducted in two stages, in which control tests were conducted and raw stormwater indicators were evaluated. Tests were carried out with time intervals, with contact durations of 30 min at a temperature 18–20 °C and repeated three times, and the final values were determined as the arithmetic means. A contact time of 30 min is the optimal detention time for the main reactions.

In Stage 1, *leaching tests* were performed to evaluate the impact of different amounts of disinfectants on stormwater pollution indicators. In the laboratory, six 1000 mL stormwater samples were prepared: one jar sample without disinfectant in order to determine water quality and five stormwater jars samples with different amounts of disinfectants, i.e., 10, 20, 30, 40 and 50 mL. These quantities were chosen to investigate low quantities as well as their associations with surface disinfections. By quantifying disinfectant volumes, including chlorine concentrations, admixture conductivity and disinfection by-products, we found that mixing is a crucial factor driving the changes in disinfectant contents and fluctuations in the conductivity of stormwater. Prepared samples were placed in a mixer and admixed for 30 min at a speed of 120 rpm (optimal mixing conditions obtained after preliminary experiment). Then, stormwater pollution indicators (pH, conductivity, turbidity, color intensity and active chlorine) were measured. Following the tests, stormwater samples were placed in hermetically sealed containers and transported for testing to a certified laboratory.

In Stage 2, *filtration tests* were conducted using a filtration laboratory bench filled with ceramzite and hemp fiber and different amounts of disinfectants (didecyldimethylammonium chloride and sodium hypochlorite). The aim was to determine the impacts of the filter filler and different amounts of disinfectants on stormwater pollution indicators.

The first step in Stage 2 was a control test using a 2000 mL sample of stormwater which was poured into a glass column. After a contact period of 30 min, the sample was measured for pH, conductivity, turbidity and color intensity. Later, another sample of stormwater was poured into a different glass column (diameter of 100 mm) filled with ceramzite (Figure 1). After 30 min of contact, the sample was measured for pH, conductivity, turbidity and color intensity.

Our third experiment evaluated the impact of filter materials on pollution indicators of stormwater contaminated with disinfectants. The control test placed hemp fiber on the ceramzite layer, and the glass column was filled with the stormwater sample. After 30 min contact, the sample was measured for pH, SEL, turbidity and color. In subsequent experiments, 10 mL of disinfectants was added to the jar with stormwater, then test water was mixed and poured into the column. After 30 min contact, pH, SEL, turbidity, color and active chlorine were measured.

Following the same order, new tests were carried out by changing the disinfectants, i.e., didecyldimethylammonium chloride was replaced by sodium hypochlorite. Stormwater pollution indicators—pH, conductivity, turbidity and color intensity—were measured.

### 2.1. Characteristics of the Materials

The impacts of the disinfectants on stormwater pollution indicators were investigated using recycled materials, i.e., a ceramzite drainage layer and hemp fiber. The ceramzite drainage layer used ceramzite sand with the following characteristics: particle density—1020 kg/m^3^, sand voids—53.63%, sand soak—33.55%.

Ceramzite sand indicator and sand soak information is presented in Table 1.

When the ceramzite fraction increased, the total open porosity of the grains and the reserve of porous space and water soak also increased, while the thickness of the walls decreased (Table 1). Microstructural studies of the 2/4 fraction ceramzite sand surfaces (Figure 2) showed that the pores on the ceramzite surface were very small in diameter (from 5 µm to 60 µm) and that sintered clay minerals occupied the largest ceramzite surface area.

Further detailed studies of ceramzite pores (Figure 3) demonstrated that the inner ceramzite micropores remained open. Through such a structure of communicating pores, water enters ceramzite, yet the number of these pores is not large, and closed pores and capillaries predominate in the inner ceramzite part.

According to its technical characteristics, *hemp fiber* (Figure 4) is one of the most resistant natural products, with good insulating properties. Compared to other natural plants (linen, cotton, etc.), hemp fiber has high absorption characteristics.

This research used hemp fiber left over from the production process. This cheap waste material is readily available in Lithuania, and its use contributes to environmentally and economically sustainable solutions.

The disinfectant used for the study was *sodium hypochlorite*, as well as *didecyldimethylammonium chloride* (C22H48ClN), which is a liquid-form, broad-spectrum bactericidal and fungicidal product used for surface cleaning. This substance is recommended for use in the disinfection of hotels, restaurants and SPA areas. Its chemical formula is C22H48ClN, its molar mass is 362.08 g/mol and it has a density of 950 kg/m³. The exposure time to inactivate bacteria is 5–20 min, whereas in the case of fungal and fungicidal contamination the exposure time is 15–30 min. (This information is according to the technical specifications for the product.)

The National Center for Public Health in Lithuania specifies that quantities of 70% ethylalcohol and 0.25% didecyldimethylammonium chloride are best for surface disinfection. In the case of heavily contaminated surfaces, disinfectants containing up to 6% of didecyldimethylammonium chloride can be used. The disinfectant complies with the international quality certificate ISO 9001, although it is flammable and may cause eye irritation.

Sodium hypochlorite is a yellow-colored, pungent-smelling, water-soluble disinfectant liquid. Due to the active chlorine, it is generally used for cleaning and disinfecting various surfaces, swimming pools, etc. Lithuania’s Ministry of Health recommends the use of 0.1% sodium hypochlorite (1:50 dilution if using household bleach with an initial concentration of 5%) for surface cleaning after cleaning with a neutral detergent. The formula for sodium hypochlorite is NaClO: molar mass—74.44 g/mol, density—1.11 g/cm³, melting point—18 °C, boiling point—101 °C. Whilst the disinfectant complies with ISO 9001, it is still harmful to the environment, irritates the respiratory tract, is highly toxic to aquatic organisms and releases toxic gases when it comes into contact with acid. (Information according to the technical specifications for the product.)

### 2.2. Measurement of Stormwater Pollution Indicators

The stormwater pollution indicators pH, conductivity, turbidity and color intensity were measured using calibrated equipment in the laboratory of Vilnius Gediminas Technical University: pH was measured with a WTW pH meter 330i, conductivity with a WTW conductivity meter 315i, turbidity and color intensity with a Thermo Scientific Genesys 10S UV and active chlorine with a CL200 ExStik. For the measurement of stormwater turbidity, the proposed method of spectrophotometry has higher accuracy than turbidimetry.

## 3. Results and Discussion

Water samples in the potentially contaminated area (Trakai, Vilnius, Lithuania) were taken at different stormwater outlets in October 2021. In order to determine the main elements in the stormwater, the collected samples were analyzed using an X-Ray analyzer (Table 2). Following the WHO’s “Cleaning and disinfection of environmental surfaces in the context of COVID-19 Interim guidance” issued on 15 May 2020, the recommendation of 0.1% (1000 ppm) in the context of COVID-19 is a conservative concentration [4]. Residual concentrations were measured after carefully disinfected surfaces were flashed with additional amounts of water; all results are presented in Table 2, Table 3, Table 4 and Table 5. All related metal and silicon concentrations were measured on special request from professional teams of disinfecting units because of the deepest concerns about possible corrosion and other related issues.

The determined active chlorine concentrations at different stormwater outlets varied from 0.0181 to 0.0313 percent. This shows that outdoor surface disinfection causes active chlorine entry into the environment. The processes of formation of different compounds of metals with chlorine and silicon matter is diverse and probably dominated by different activities, such as adsorption, ion exchange and chelation, under different circumstances.

The silicon concentrations obtained for the investigated water samples showed that some of the chlorine was formed on the surface of sand particles. Silicon concentrations varied from 1.944 to 2.4 percent.

The following elemental concentrations were obtained: nickel (Ni)—0.00212 percent, copper (Cu)—0.00593 percent, zinc (Zn)—0.00623 percent, chromium (Cr)—0.0162 percent. An assumption was made that disinfectants based on chlorine affect the anti-corrosion coatings of chromium, nickel, copper and zinc (Table 4).

Conventional oxidation is a global process involving the use of chlorine compounds, in which electrons are removed from a surface to increase its oxidation level. Disinfection destructs some organics into more simple substances, making them easier to remove by natural reactions. The major components and characteristics of contaminants are colloidal, dissolved organic matter and suspended solids.

Leaching tests were performed to evaluate the disinfectants’ impacts on stormwater pollution indicators. See Table 5 for the stormwater pollution indicators of the samples investigated in the laboratory.

Depending on the amounts of disinfectant (10–50 mL) present in the water samples, the pH indicators varied from 6.731 to 7.315, the conductivity varied slightly from 273 to 259 µS/cm and color intensity and turbidity varied, respectively, within the ranges of 2.87–3.00 AV and 0.087–1.926 NTU.

A rapid increase in turbidity values was observed for the sample with 20 mL of disinfectant, and in most of the experiments the highest turbidity values were observed in the first runs. The results showed that the removal of turbidity-causing materials occurs faster than the removal of colored materials; however, no increase in color value was observed after additional doses of disinfectant were added.

When different amounts of disinfectants were added to the investigated stormwater samples, changes in stormwater pollution indicators were obtained. Figure 5 presents the increases in sample water pH with increasing amounts of disinfectants (10–50 mL), which varied in the range of 6.8–7.5. The results indicated that many influences, such as hemp fibers, the concentration of disinfectant, the experimental run and the proportion of disinfectant admixed, as well as inner phase acidity, the ratio of ceramzite inner and the ratio of stormwater, contributed to the effectiveness of filtration.

The first run investigated water sample conductivity depending on disinfectant amounts, which varied slightly in the range of 250–270 µS/cm. A replicate run obtained a significant increase in conductivity with increasing disinfectant amounts (10–50 mL). The highest value of 1000 µS/cm was fixed when the investigated sample water was contaminated with 50 mL of disinfectant (Figure 5). Stormwater filtration was investigated in stabilization runs of replicates for the column experiment. The results showed that the presence of soluble matter in the filter media affected the filtration process and increased the conductivity in replicate runs. It was observed that the presence of filter media substances increased the process of filtration and resulted in a reduction in conductivity in the first run of the experiment.

Figure 6 presents the changes in the investigated water samples’ color intensity and turbidity indicators depending on the disinfectant amounts. An increase in the investigated samples’ water turbidity was obtained from 0.067 to 1.93 NTU. The highest turbidity value for the investigated water sample was fixed when the sample was contaminated with 50 mL of disinfectant.

When the investigated water samples’ contamination with disinfectants increased, color intensity indicators increased from 0.3 to 3 AV as well. A replicate run showed that color intensity increased (2.4) when 10 mL of disinfectant was added to the sample and that it decreased to 1.5 AV with the addition of 50 mL of disinfectant. A second leaching test showed that by increasing the ceramzite amount in the investigated water sample contaminated with disinfectants, the conductivity indicator increased with increasing ceramzite and disinfectant amounts. Figure 7 shows that the investigated water sample conductivity increased from 425 to 442 µS/cm when the water was contaminated with 40 mL and 50 mL of disinfectant. When the amounts of disinfectant were 20 and 30 mL, conductivity values were in the range of 417–416 µS/cm.

It can be seen in Figure 8 that the pH indicator changed slightly (from 7.81 to 7.85), that turbidity change varied in the range of 0.070–0.145 NTU and that the highest value of color intensity (1.932 AV) was obtained when 50 mL of disinfectant was added to the investigated sample water.

In a continuing filtration of the investigated sample water contaminated with disinfectants, hemp fiber was added on top of the ceramzite layer. Changes in stormwater pollution indicator values (pH, conductivity and turbidity) before and after filtration were monitored. The adsorption of disinfectants onto hemp fibers supported the attainment of equilibrium only according to pH values. Adsorption and desorption runs for conductivity and stormwater color were not in equilibrium between solid and liquid phases. The process is a combination of ionic attraction and repulsion, substance bounding, ion–dipole forces, covalent bonding and internal forces. Hemp fibers exhibit extensive porosity and large internal surface areas, which features enhance the accessibility of cations to carboxylic and other functional groups.

Figure 9 presents disinfectant impacts on pH indicators.

The pH indicator of the investigated sample water varied in the range of 6.5–8.0. Before filtration, the pH indicator was 6.5–7.0. The real pH values were affected by the optimal conditions under which the filter media were used, e.g., disinfectant–stormwater ratio and the time and temperature of the filtered stormwater. In natural aqueous systems, the pH levels of the filtered stormwater showed the greatest pH effects in the mixtures obtained. The levels of such changes depended on the specific filter media–filtrate system that was used. In the case of commercially produced hemp fibers, the pH was due to the presence of organic matter either present in the natural material or introduced in the production processes. Subsequent to filtration, most hemp fibers were slightly alkaline; washing and drying processes may change pH values.

Figure 9 shows a slight change in conductivity in the range of 80–100 µS/cm obtained before filtration. The first and second samples showed an increase in conductivity after filtration, meanwhile, for the third, fourth and fifth samples, decreases in conductivity from 139 to 119 µS/cm were obtained. By comparing conductivity indicators before filtration (86.5–100.5 µS/cm), it was determined that the conductivity values were higher after filtration (429–119 µS/cm). Functional groups on the surface of filter media often participate in important reactions in solid–liquid interactions. Polar compounds may be accepted by carbonyl or hydroxyl activities; inorganic materials may be bonded by amine groups and nonpolar disinfectants bound by surface nonpolar filter media.

In the laboratory, before carrying out the assessment of the impact of sodium hypochlorite on stormwater pollution indicators, the indicators of the sample water taken from the possibly contaminated area and the indicators of the same sample water after its filtration using hemp fibers were measured.

The obtained results showed that the pH of the investigated sample water was higher (7.82) than its value after filtration (7.47), that conductivity increased from 86.7 to 95.5 µs/cm and that the turbidity and color intensity indicators were 0.097–0.153 NTU and 1.109–1.119 NTU.

After contamination of the investigated sample water with 10 mL of sodium hypochlorite, stormwater pollution indicators were measured before and after filtration. Table 6 presents the decreases in pH and conductivity of the investigated sample water. The pH before filtration was 11.6; after filtration it was 1.36. Conductivity before filtration was 5.6; after filtration it was 5.0 µs/cm. The explanations of the changes in pH and other indicators might be as follows. The research results showed that, after filtration, conductivity values were higher by two to five times. For instance, increases in pH indicators were obtained after filtration, with pH levels in the range of 7.0–8.0 (pH indicator values before filtration were 6.5–7.0). It might be that the increase in pH was due to the alkalinity that stormwater acquires when it comes into contact with ceramzite and disinfectants. Changes in conductivity might also be caused by disinfectant amounts. The research showed that conductivity increases with higher amounts of disinfectants. Dissolved disinfectants conduct electrical current, and conductivity increases depending on disinfectant amount and contact with filtering materials. An increase in the turbidity of the investigated sample water was obtained from 0.067 to 1.93 NTU. The highest turbidity value for the investigated sample water was obtained when the sample water was contaminated with a 50 mL amount of disinfectant. It might be assumed that the intensity of turbidity was impacted by water sample contact with the filtering materials. When the investigated sample water’s contamination with disinfectants increased, the color intensity indicator increased from 0.3 to 3 AV as well. A replicate run showed that color intensity increased (2.4) when 10 mL of disinfectant was added to the sample water and that it decreased to 1.5 AV with 50 mL of disinfectant. It might be assumed that the disinfectant diluted the water sample and decreased the color of the sample water.

Increases in color intensity and turbidity were obtained, and the amount of active chlorine decreased from 2.93 ppm to 1.0 ppm. In order to find out whether active chlorine remained after filtration, the column was rinsed with the investigated sample water and the stormwater pollution indicators were measured.

After the measurements, the active chlorine amount was fixed below the limit of <0.01 ppm. This preliminary research shows that disinfection might cause active chlorine entry into the environment. According to the latest scientific information, active chlorine in the environment [24] should be monitored and detected.

## 4. Conclusions


After leaching and filtration tests, it was established that recycled materials (hemp fibers) could be used to remove disinfectants from stormwater. By evaluating the research results and analyzing possible ways of removing disinfectants from stormwater, the removal efficiency obtained was approximately 66 percent.To determine the impact of disinfection on stormwater pollution indicators using different recycled materials (ceramzite, hemp fibers) and different amounts of disinfectants, leaching and filtration tests are advised. After filtration, conductivity values were two to five times higher.The research on stormwater contaminated with disinfectants showed that when the amount of disinfectant changed, the stormwater pollution indicators (pH, conductivity and turbidity) also changed. Although the investigations have shown this, more studies are needed to explain these processes. The assumption made here is that a chemical reaction is taking place and that the pollutants settle or form flakes.The analyses of the sample water contaminated with sodium hypochlorite showed that disinfection might cause active chlorine to enter the environment such that it is necessary to monitor and detect chlorine in the environment and apply possible methods for its removal. The investigations have shown that, after filtration, initial concentrations of chlorine decreased to a detection limit below 0.01.This research has shown that in order to obtain more accurate results it is necessary to compare research performed in situ and ex situ. Differences in results may be caused by climatic conditions, the contact time of the waste materials with the investigated sample water and other conditions. Therefore, it is necessary to continue testing in field conditions.


## Figures and Tables

**Figure 1 ijerph-19-14486-f001:**
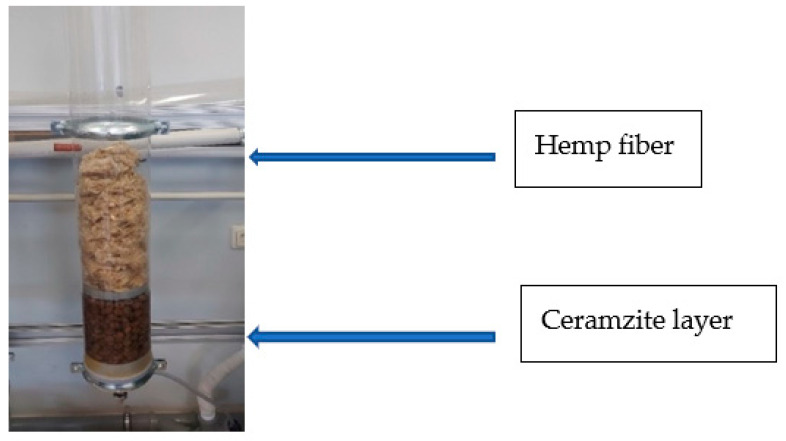
Glass column filled with a ceramzite layer and hemp fiber.

**Figure 2 ijerph-19-14486-f002:**
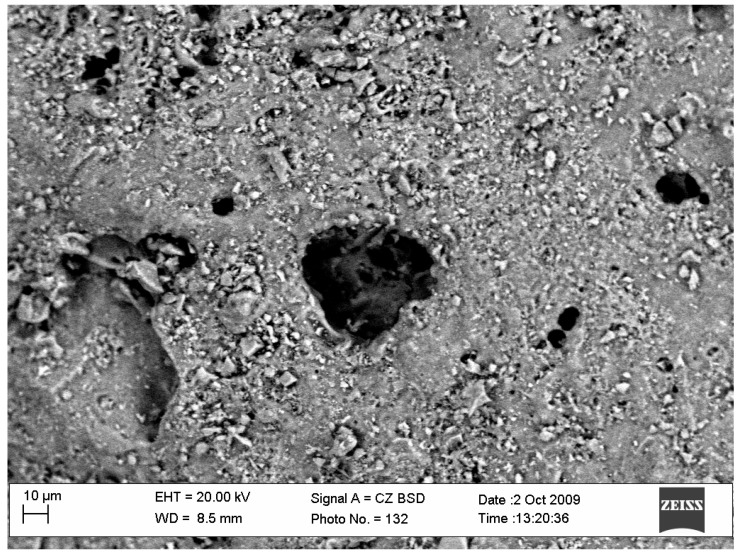
Microstructure of a ceramzite granule.

**Figure 3 ijerph-19-14486-f003:**
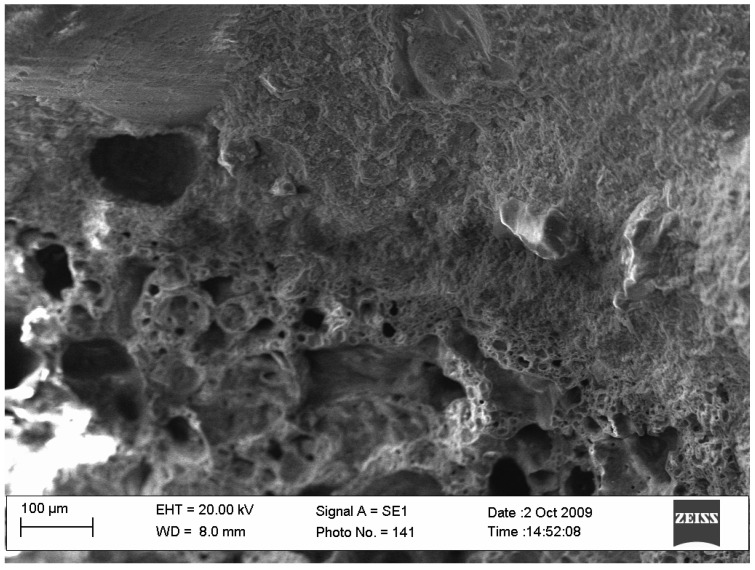
Ceramzite sand and pore microstructure.

**Figure 4 ijerph-19-14486-f004:**
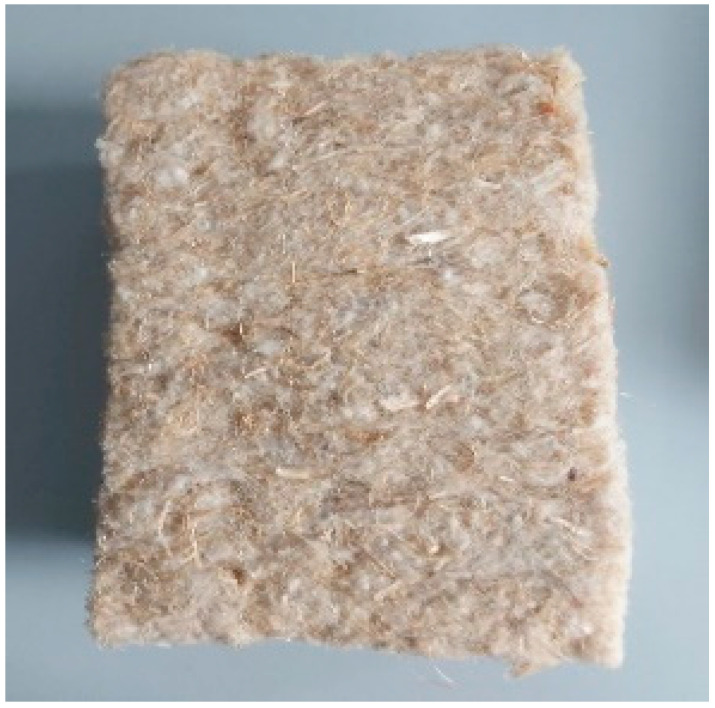
Hemp fiber.

**Figure 5 ijerph-19-14486-f005:**
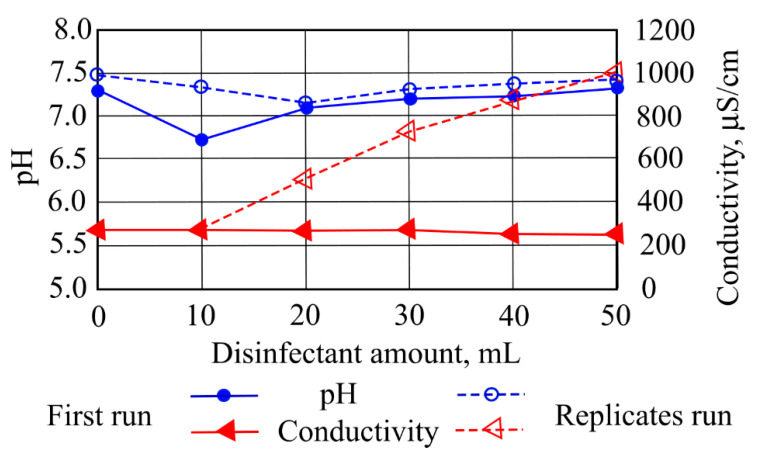
The impact of disinfectants on pH and conductivity.

**Figure 6 ijerph-19-14486-f006:**
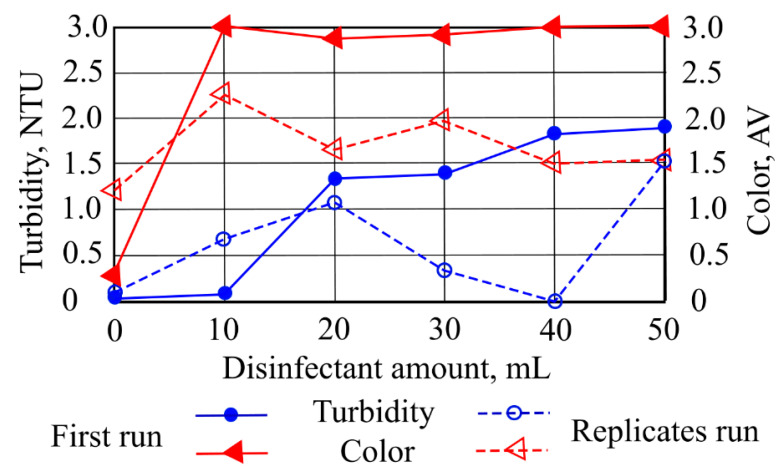
The impact of disinfectants on turbidity and color intensity.

**Figure 7 ijerph-19-14486-f007:**
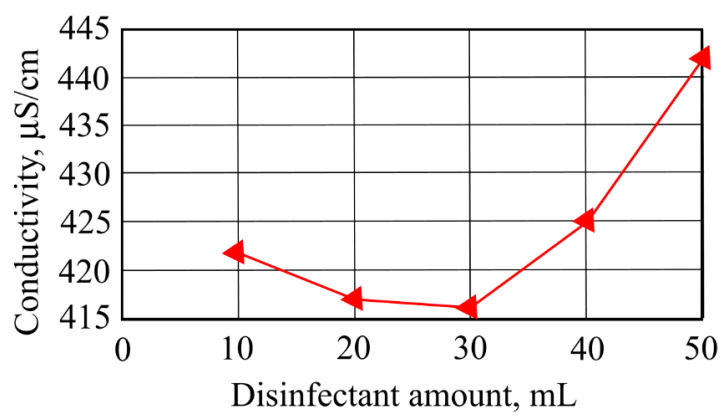
The impact of disinfectants on conductivity.

**Figure 8 ijerph-19-14486-f008:**
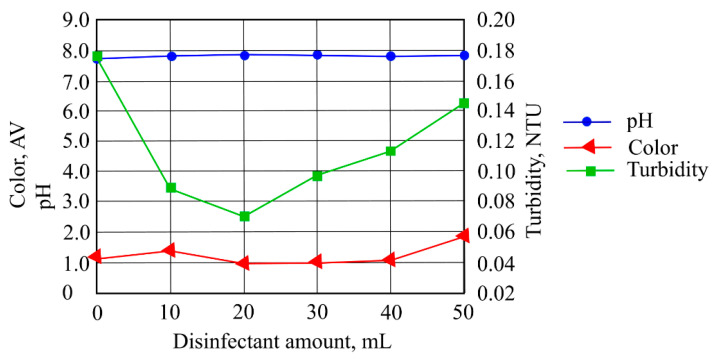
The impact of disinfectants on stormwater pollution indicators after contact with waste materials.

**Figure 9 ijerph-19-14486-f009:**
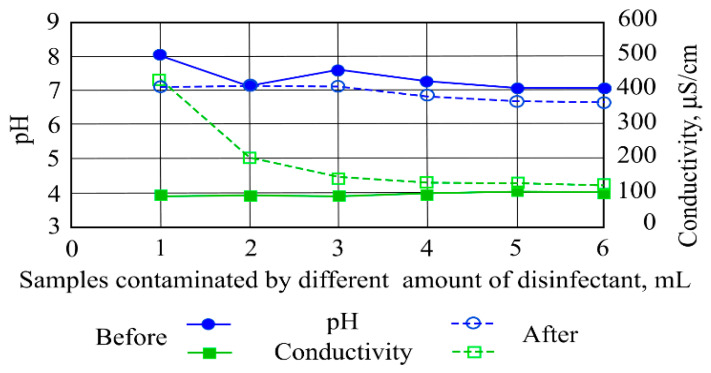
The impacts of disinfectants on stormwater pollution indicators before and after filtration.

**Table 1 ijerph-19-14486-t001:** Ceramzite sand indicator and sand soak information.

Ceramzite Fraction	W_R_, %	W_E_, %	D, %	R, %	W, %
2/4	62.92	26.92	0.59	57.22	42.9
4/10	65.69	23.81	0.52	63.76	44.7
10/20	74.09	25.67	0.35	65.35	56.9

Abbreviations: W_R_—total open porosity, W_E_—effective porosity, D—conditional thickness of pore and capillary walls, R—porous space reserve, W—water soak after 72 h.

**Table 2 ijerph-19-14486-t002:** Elements obtained in stormwater samples (concentration, %) (Antaviliai, Vilnius Region).

Element	Sample 1	Sample 2	Sample 3	Sample 4	Sample 5	Sample 6
Na	<0.010	<0.010	<0.037	<0.010	<0.010	<0.035
Si	3.214	3.010	2.342	2.215	2.670	2.661
Cl	0.02393	0.02298	0.03130	0.02945	0.02020	0.01811
K	0.0611	0.0532	0.0244	0.0283	0.0424	0.0425
Ca	4.508	4.575	4.951	4.814	4.607	4.506
Cr	0.00801	0.00861	0.00910	0.0848	<0.00010	0.00822
Ni	0.00167	0.00115	0.00199	0.00194	0.00137	0.00148
Cu	0.00246	0.00247	0.00206	0.00243	0.00201	0.00216
Zn	0.00959	0.01027	0.01226	0.01135	0.00308	0.00307

**Table 3 ijerph-19-14486-t003:** Elements obtained in stormwater samples (concentration, %) (Paneriai, Vilnius Region).

Element	Sample 1	Sample 2	Sample 3	Sample 4	Sample 5	Sample 6
Na	<0.010	<0.036	0.0379	<0.025	0.0559	<0.036
Si	2.372	2.400	2.372	2.322	1.944	2.400
Cl	0.01471	0.01673	0.04232	0.03916	0.03581	0.01673
K	0.0282	0.0297	0.0778	0.0728	0.0668	0.0297
Ca	4.603	4.791	1.463	1.366	1.162	4.791
Cr	0.00768	0.00844	0.0162	0.0133	0.0144	0.00844
Ni	0.00127	0.00147	0.00075	<0.00073	0.00092	0.00147
Cu	0.00140	0.00175	0.00499	0.00593	0.00539	0.00175
Zn	0.00252	0.00273	0.00494	0.00505	0.00550	0.00273

**Table 4 ijerph-19-14486-t004:** Elements obtained in stormwater samples (concentration, %) (Trakai City, Vilnius Region).

Element	Sample 1	Sample 2	Sample 3	Sample 4	Sample 5	Sample 6
Na	0.0379	<0.025	0.0559	0.0412	0.0807	0.0792
Si	2.372	2.322	1.944	2.309	8.193	6.953
Cl	0.04232	0.03916	0.03581	0.04226	0.02635	0.02548
K	0.0778	0.0728	0.0668	0.0815	0.4222	0.3298
Ca	1.463	1.366	1.162	1.298	4.236	3.825
Cr	0.0162	0.0133	0.0144	0.0142	0.01378	0.00958
Ni	0.00075	<0.00073	0.00092	0.00081	0.00212	0.00226
Cu	0.00499	0.00593	0.00539	0.00508	0.00417	0.00398
Zn	0.00494	0.00505	0.00550	0.00504	0.00499	0.00623

**Table 5 ijerph-19-14486-t005:** The impact of disinfectants on stormwater pollution indicators.

Disinfectant Amount	Empty without Disinfectant	10 mL	20 mL	30 mL	40 mL	50 mL
Stormwater pollution indicators: pH	7.30	6.731	7.091	7.194	7.229	7.315
Conductivity: µS/cm	273	273	273	276	258	259
Turbidity: NTU	0.067	0.087	1.36	1.42	1.837	1.926
Color intensity: AV	0.27	3.00	2.87	2.92	3.00	3.00

**Table 6 ijerph-19-14486-t006:** Pollution indicators of sample water contaminated with sodium hypochlorite.

	Sample Water	Sample Water after Filtration	Sample Water with Disinfectant after/before Filtration	Sample Water with Disinfectant	Sample Water without Disinfectants
pH	7.820	7.468	11.6/11.36	11.72	10.671
Conductivity, µs/cm	86.7	95.5	5.6/5.0	10.47	2.38
Turbidity, NTU	0.097	0.153	3.0/0.220	0.193	0.283
Color intensity, AV	1.109	1.192	3.0/1.394	1.477	1.639
CL, ppm			2.93/1.00	3.60	<0.01
Disinfectant amount, mL			10.0/10.0	10.0	<0.01

## Data Availability

The data presented in this study are available on request from the corresponding author.

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
