# Peer review of "Experimental Research on the Treatment of Stormwater Contaminated by Disinfectants Using Recycled Materials—Hemp Fiber and Ceramzite"

_ijerph, 2022, doi:10.3390/ijerph192114486_

Round 1

Reviewer 1 Report

The manuscript titled "Experimental research on the treatment of stormwater contaminated by disinfectants using recycled materials - hemp fiber and ceramzite", reports interesting changes in stormwater indicators due to the presence of disinfectants. The authors also reported a novel, eco-friendly remediation measure of hemp-ceramzite filter that can be used to remove the disinfectants. A point of concern in this manuscript is that the authors did not take the effort in discussing the results reported. The authors only listed the results obtained but did not provide any explanation as why the pH changed, why the conductivity increased after filtration etc. It is necessary to explain the results observed

Author Response

Review 1

Open Review

(x) I would not like to sign my review report

( ) I would like to sign my review report

English language and style

( ) Extensive editing of English language and style required

( ) Moderate English changes required

(x) English language and style are fine/minor spell check required

( ) I don't feel qualified to judge about the English language and style

Yes         Can be improved              Must be improved           Not applicable

Does the introduction provide sufficient background and include all relevant references?

( )            (x)          ( )            ( )

Are all the cited references relevant to the research?

(x)          ( )            ( )            ( )

Is the research design appropriate?

( )            (x)          ( )            ( )

Are the methods adequately described?

(x)          ( )            ( )            ( )

Are the results clearly presented?

( )            ( )            (x)          ( )

Are the conclusions supported by the results?

( )            ( )            (x)          ( )

Comments and Suggestions for Authors

The manuscript titled "Experimental research on the treatment of stormwater contaminated by disinfectants using recycled materials - hemp fiber and ceramzite", reports interesting changes in stormwater indicators due to the presence of disinfectants. The authors also reported a novel, eco-friendly remediation measure of hemp-ceramzite filter that can be used to remove the disinfectants. A point of concern in this manuscript is that the authors did not take the effort in discussing the results reported. The authors only listed the results obtained but did not provide any explanation as why the pH changed, why the conductivity increased after filtration etc. It is necessary to explain the results observed

Dear Reviewer,

Thank you very much for your valuable suggestions.

The explanations on the changes of pH and other indicators might be these: research results showed after filtration conductivity values are higher 2-5 times. For instance increase of pH indicator was obtained after filtration, pH was in the limits of po 7.0 -8.0 (pH indicator values before filtration pH indicator was 6.5 – 7.0). It might be explained that the increase of pH is due to the alkalinity that stormwater picks up when coming into contact with ceramzite and disinfectants. Changes of conductivity might be also caused by disinfectants amount. Research showed that by higher amount of disinfectants conductivity increases. Dissolved disinfectants conduct electrical current and conductivity increases depending on disinfectants amount and contact with filtering materials. An increase of investigated sample water turbidity  was obtained from 0.067 to 1.93 NUT. The highest turbidity value of investigated sample water was fixed when sample water was contaminated with 50 ml. amount of disinfectant. It might be assumed that intensity of turbidity  is impact by water sample contact with filtering materials. When investigated sample water contamination by disinfectants increases, color intensity indicator increases from 0.3 to 3 AV as well. A replicate run shows color intensity increase (2.4) when 10 ml was added to sample water and it decrease to 1,5 AV at 50 ml of disinfectant. It might be assumed that disfectants dilute water sample and decrease color of sample water.  In general, investigations have shown that, more studies are needed to explain these processes. The assumption to be made here is that a chemical reaction is taking place and the pollutants settle or form flakes.

There were changes and detailed explanations added step by step according to all Reviewers comments and suggestions in track change format.

We express our gratitude to Reviewers comments,

Sincerely on behalf of coauthors Professor Dr Marina Valentukeviciene.

Reviewer 2 Report

- English should be revised.

- Additional numerical data of the core findings should be added to the abstract to be more informative.

- Pronouns should not use in the text.

- The origination of COVID-19 should not mention in the text respecting Wuhan and its people. 

- What is the rule for selecting the following: "contact duration 30 min at a temperature 18 - 20 ºC", "different amounts of disinfectants, 100 i.e. 10, 20, 30, 40, 50 ml.", "speed of 120 rpm"?

- You should list an additional column in Table 2 showing permission limits according to WHO.

- What is the reason for the wide difference in the conductivity values between the first run and the replicates run? The same thing is observed for other indicators. Please explain.

- More interpretation is required to explain the irregular behavior noted in Figures.

- Why does the pH value change after filtration? 

Author Response

Review 2

Open Review

( ) I would not like to sign my review report

(x) I would like to sign my review report

English language and style

( ) Extensive editing of English language and style required

( ) Moderate English changes required

(x) English language and style are fine/minor spell check required

( ) I don't feel qualified to judge about the English language and style

Yes         Can be improved              Must be improved           Not applicable

Does the introduction provide sufficient background and include all relevant references?

( )            (x)          ( )            ( )

Are all the cited references relevant to the research?

(x)          ( )            ( )            ( )

Is the research design appropriate?

( )            (x)          ( )            ( )

Are the methods adequately described?

( )            (x)          ( )            ( )

Are the results clearly presented?

( )            ( )            (x)          ( )

Are the conclusions supported by the results?

( )            (x)          ( )            ( )

Comments and Suggestions for Authors

Dear Reviewer,

Thank you for the comments and suggestions provided in your review. Please find below explanations and answer to the comments

- English should be revised

Considering the given suggestion text has been revised by the experienced English speaking editor.

- Additional numerical data of the core findings should be added to the abstract to be more informative.

Considering the comment conclusions and abstract has been corrected.

  1. Conclusions were added
  2. After leaching and filtration tests, it was established that recycled materials (hemp fiber) could be used to remove disinfectants from stormwater. By evaluating research results and analyzing possible ways to remove disinfectants from stormwater, the removal efficiency obtained was approximately 66 percent.
  3. To determine the impact of disinfection on stormwater pollution indicators using different recycled materials (ceramzite, hemp fiber), and different amounts of disinfectants, leaching and filtration tests are advised. After filtration conductivity values are higher 2-5 times. Increase of pH indicator was obtained after filtration, pH was in the limits of po 7.0 -8.0 (pH indicator values before filtration pH indicator was 6.5 – 7.0).
  4. Research on stormwater contaminated by disinfectants show that when the amount of disinfectant changes, stormwater pollution indicators (pH, conductivity, turbidity) also change. Investigations have shown that, more studies are needed to explain these processes. The assumption to be made here is that a chemical reaction is taking place and the pollutants settle or form flakes.
  5. Sample water contaminated by sodium hypochlorite research shows that disinfection might cause active chlorine enter to the environment and it is necessary to monitor, detect and apply possible methods of chlorine removal from environment. Investigations have shown that after filtration initial concentrations of chlorine are decreased till detection limit below 0.01

4.5. This research showed that in order to obtain more accurate results it is necessary to compare research performed in situ and ex situ. Differences in results may be caused by climatic conditions, contact time of the waste materials and investigated sample water and other conditions. Therefore, it is necessary to continue testing in field conditions.

- Pronouns should not use in the text.

Changed by professional editing specialist.

- The origination of COVID-19 should not mention in the text respecting Wuhan and its people.

We apologizes for mentioning Wuhan region in the text. Introduction part was changed accordingly (lines 29 and 36):

....During tThe  COVID-19 pandemic started which is believed to have originated in the city of Wuhan in China at the end of 2019 saw an increase in the use of disinfectants to clean outdoor spaces. In an attempt to arrest, and possibly halt, the spread of contagion during the pandemic, streets, highways and other public spaces were sprayed using various chemical substances, which led to increased environmental pollution. One estimate claims that from the start of the pandemic until March 2020 more than 200 tonnes of disinfectants were used in some regions  Wuhan alone [1]. Furthermore, during February and March 2020 amounts of chlorine were found to be present in investigated Chinese lakes [2].

- What is the rule for selecting the following: "contact duration 30 min at a temperature 18 - 20 ºC", "different amounts of disinfectants, 100 i.e. 10, 20, 30, 40, 50 ml.", "speed of 120 rpm"?

Additional explanation added to methodical part.

- You should list an additional column in Table 2 showing permission limits according to WHO.

There are WHO reference added to the text before Table 2 with disinfectants limits and detailed explanation.

- What is the reason for the wide difference in the conductivity values between the first run and the replicates run? The same thing is observed for other indicators. Please explain.

Explanation is added to every characteristic following appearance in the text in trach change format.

- More interpretation is required to explain the irregular behavior noted in Figures.

Detailed interpretations were added to the main text after related Figures.

- Why does the pH value change after filtration?

The reason is explained in the text. The real value of pH was affected by the optimal conditions under which filter media used, e.g. disinfectant-stormwater ratio and the time and temperature of the filtered stormwater. With natural aqueous systems, the pH of the filtered stormwater shows its highest effect on the pH of the mixture being obtained. The level of such charges depends on the specific filter media-filtrate system being used. In the case of commercial produced fiber hemp, the pH is due to the presence of organic matter either present in the natural material or included during production processes. Subsequent to filtration most hemp fibers are slightly alkaline; washing and drying processes may change pH values.

There were changes and detailed explanations added step by step according to all Reviewers comments and suggestions in track change format.

We express our gratitude to Reviewers comments,

Sincerely on behalf of coauthors Professor Dr Marina Valentukeviciene.

Reviewer 3 Report

This work investigated the impact of disinfectants on stormwater using the waste materials ceramzite and hemp fiber. However, the following issues still need to be clarified.

1. It is suggested that the author needs to refine the abstract section. The authors even did not mention the results of research about the impact of disinfectants on stormwater.

2. In line 68, why the authors cited 8 references in this sentence?

3. In line 95, the unit of temperature is incorrect. In addition in line 310 and 330, it is 1.5 AV, 7.81, 7.85, not 1,5 AV, 7,81, 7,85.

4. Why does the author use ultraviolet spectrophotometer to measure turbidity of samples instead of turbidimeter?

Author Response

Review 3

Open Review

(x) I would not like to sign my review report

( ) I would like to sign my review report

English language and style

( ) Extensive editing of English language and style required

( ) Moderate English changes required

(x) English language and style are fine/minor spell check required

( ) I don't feel qualified to judge about the English language and style

Yes         Can be improved              Must be improved           Not applicable

Does the introduction provide sufficient background and include all relevant references?

(x)          ( )            ( )            ( )

Are all the cited references relevant to the research?

(x)          ( )            ( )            ( )

Is the research design appropriate?

(x)          ( )            ( )            ( )

Are the methods adequately described?

(x)          ( )            ( )            ( )

Are the results clearly presented?

(x)          ( )            ( )            ( )

Are the conclusions supported by the results?

( )            (x)          ( )            ( )

Comments and Suggestions for Authors

This work investigated the impact of disinfectants on stormwater using the waste materials ceramzite and hemp fiber. However, the following issues still need to be clarified.

  1. It is suggested that the author needs to refine the abstract section. The authors even did not mention the results of research about the impact of disinfectants on stormwater.

Dear Reviewer,

Abstract section was added with mentioned results.

The laboratory tests conducted filtered stormwater samples contaminated by different amounts of disinfectants using hemp fiber and ceramzite with the amount of active chlorine decrease from 2.93 ppm till 1.0 ppm.  Changes in pH levels, conductivity, turbidity and color intensity were monitored before and after filtration, pH indicator changes slightly (from 7.81 to 7.85), turbidity changes varies in the limits of 0.070-0.145 NTU, the highest value of color intensity (1.932 AV) was obtained when 50 ml of disinfectant was added to investigated sample water.  

  1. In line 68, why the authors cited 8 references in this sentence?

When chlorine compounds enter stormwater they react with the organic, inorganic and anthropogenic pollutants present to form a variety of by-products such as haloacetic acids, trihalomethanes, dibromochloromethane, bromodichloromethane, tribromoethane, dichloroacetonitrile, chloramines, and other compounds [ 17, 20]. These toxic compounds effects aquatic ecosystems and may also have various side effects on the water environment [15, 16,18,]. For example increased chlorine concentration can cause

chlorine toxicity in plants [14] and ecological imbalance [19].Chlorine by-products accumulate on plankton and fish in the aquatic environment and can cause negative chronic effects on aquatic organisms [21].

  1. In line 95, the unit of temperature is incorrect. In addition in line 310 and 330, it is 1.5 AV, 7.81, 7.85, not 1,5 AV, 7,81, 7,85.

Line 95, 310 and 330 were corrected according by the comments.

Line 95: Tests were carried out according to the time intervals by contact duration 30 min at a temperature 18 - 20 oC and repeated three times, and the final value determined as the arithmetic mean

Line 310: When investigated sample water contamination by disinfectants increases, color intensity indicator increases from 0.3 to 3 AV as well. A replicate run shows color intensity increase (2.4) when 10 ml was added to sample water and it decrease to 1.5 AV at 50 ml of disinfectant.

Line 330: See Figure 8 pH indicator changes slightly (from 7.81 to 7.85), turbidity changes varies in the limits of 0.070-0.145 NUT, the highest value of color intensity (1.932 AV) was obtained when 50 ml of disinfectant was added to investigated sample water.

  1. Why does the author use ultraviolet spectrophotometer to measure turbidity of samples instead of turbidimeter?

For the measurement of a stormwater turbidity, the proposed method of spectrophotometry has higher accuracy than turbidimeter.

There were changes and detailed explanations added step by step according to all Reviewers comments and suggestions in track change format.

We express our gratitude to Reviewers comments,

Sincerely on behalf of coauthors Professor Dr Marina Valentukeviciene.

Round 2

Reviewer 1 Report

The authors have addressed the concerns that have been raised

Reviewer 2 Report

None